# IL-10-Directed Cancer Immunotherapy: Preclinical Advances, Clinical Insights, and Future Perspectives

**DOI:** 10.3390/cancers17061012

**Published:** 2025-03-17

**Authors:** Adel G. El-Shemi, Afnan Alqurashi, Jihan Abdullah Abdulrahman, Hanin Dhaifallah Alzahrani, Khawlah Saad Almwalad, Hadeel Hisham Felfilan, Wahaj Saud Alomiri, Jana Ahmed Aloufi, Ghadeer Hassn Madkhali, Sarah Adel Maqliyah, Jood Bandar Alshahrani, Huda Taj Kamal, Sawsan Hazim Daghistani, Bassem Refaat, Faisal Minshawi

**Affiliations:** 1Department of Clinical Laboratory Sciences, Faculty of Applied Medical Sciences, Umm Al-Qura University, Al Abdeyah, Makkah 21955, Saudi Arabia; agshemi@uqu.edu.sa (A.G.E.-S.); jihanasalama.21@gmail.com (J.A.A.); haneendhaif.mls@gmail.com (H.D.A.); khawlhsaad@gmail.com (K.S.A.); hadeel.h.filfilan@gmail.com (H.H.F.); whajalomiri2001@gmail.com (W.S.A.); janaaloufii@gmail.com (J.A.A.); ghadeermadkhali@gmail.com (G.H.M.); maqliyahsarah@gmail.com (S.A.M.); joodbandarr@gmail.com (J.B.A.); hudatajkamal@gmail.com (H.T.K.); s-dagh@hotmail.com (S.H.D.); barefaat@uqu.edu.sa (B.R.); 2Department of Pharmacology, Faculty of Medicine, Assiut University, Assiut 71515, Egypt; 3Independent Researcher, Makkah 21955, Saudi Arabia; dr.fona1101@gmail.com; 4Department of Hematology, Dr. Sulaiman Al-Habib Medical Diagnostic Laboratory, Olaya District, Riyadh 12234-3785, Saudi Arabia; 5Department of Blood Bank and Laboratory, Saudi German Hospital, Makkah 24242, Saudi Arabia

**Keywords:** interleukin 10, cancer immunotherapy, pegylated IL-10, IL-10 fusion proteins and delivery systems, engineered IL-10 variants

## Abstract

Interleukin-10 (IL-10) is emerging as a promising cancer immunotherapy due to its ability to boost exhausted immune cells (CD8^+^ T cells), helping them fight tumors more effectively while also reducing harmful inflammation. However, despite encouraging early research, clinical trials of IL-10-based treatments, such as pegilodecakin, have faced challenges like side effects and inconsistent results, highlighting the need for better dosing and delivery methods. To overcome these obstacles, scientists have developed engineered IL-10 variants, fusion proteins, and bispecific constructs, which improve stability and extend IL-10’s activity, enhancing its effects when used with immune checkpoint inhibitors. As research continues to refine IL-10 therapies, innovative delivery systems and combination treatments could unlock its full potential in precision cancer care, making IL-10 a key player in the future of immunotherapy.

## 1. Introduction

Interleukin-10 (IL-10) is a dimeric cytokine encoded by the *IL-10* gene on chromosome 1 [1]. It is produced by a diverse array of cells, including immune cells such as CD4^+^ T cells (Th1, Th2, and regulatory T (Tregs) cells), myeloid-derived suppressor cells (MDSCs), natural killer (NK) cells, mast cells, B cells, CD8^+^ T cells, and dendritic cells (DCs) [2,3,4]. Additionally, IL-10 is secreted by tumor cells [5], further emphasizing its broad cellular origin. Mechanistically, IL-10 exerts its effects through a heterodimer receptor complex composed of two IL-10Rα (IL-10R1) subunits and two IL-10Rβ (IL-10R2) subunits [6,7,8]. Upon receptor binding, IL-10 initiates downstream signaling cascades predominantly through Janus kinase 1 (JAK1) and tyrosine kinase 2 (Tyk2). This activation leads to the phosphorylation of signal transducer and activator of transcription (STAT) proteins, thereby modulating the expression of key target genes essential for IL-10’s anti-inflammatory and immunoregulatory functions [9,10,11,12]. IL-10Rs are expressed on various immune-associated hematopoietic cells as well as non-immune cells, such as epithelial cells, endothelial cells, and fibroblasts. However, their expression patterns vary depending on the cell type and specific biological context, and this diversity enables IL-10 to exert a wide range of signaling activities [13,14]. Moreover, the expression of IL-10Rs, particularly IL-10R1, is tightly regulated and can be upregulated in response to inflammatory stimuli or activation of specific signaling pathways [13,14].

Initially identified as a potent anti-inflammatory cytokine [15], IL-10 has since emerged as a pleiotropic molecule with diverse regulatory roles in immune responses [16,17] and cancer biology, encompassing both tumor-promoting and anti-tumor effects [18,19,20]. The key determinants of the paradox tumor-promoting and anti-tumor properties of IL-10 include tumor type, the tumor microenvironment (TME) context, IL-10 sources, concentration, and intratumoral persistence, IL-10R expression levels and binding affinities, activation of JAK/STAT downstream signaling pathways on tumor and immune cells, and the intricate interplay between IL-10 and other immunoregulatory molecules within the TME [21,22,23,24,25,26].

Given its diverse sources, multifaceted functions, and the multiple variables that can modulate its signaling activity, targeting IL-10 for cancer immunotherapy presents a formidable challenge [27]. Notably, IL-10 expression by its heterogeneous cellular subsets can be modulated at multiple levels, including chromatin remodeling, transcriptional control, post-transcriptional processing, and cellular differentiation states. While some regulatory mechanisms are conserved across cell types, others exhibit cell type- or stimulus-specific variability [28,29,30]. For instance, innate immune cells rely on pathogen-derived products, such as those recognized by pattern recognition receptors (PRRs), including Toll-like receptors (TLRs), to induce IL-10 production [31]. Modulation by autocrine or paracrine cytokines further influences IL-10 production in these innate immune cells [28]. In contrast, naïve CD4^+^ T cells require differentiation into subsets before acquiring the ability to produce IL-10 [32]. Furthermore, several signaling pathways influence IL-10 production, including MAPK/ERK1/2 [33,34], PI3K-AKT [35], NF-κB [36], and mTOR pathways [37]. Similarly, transcription factors such as c-Maf [38], Jun family members (JunB, c-Jun) [39], Foxp3 [40], aryl hydrocarbon receptor (AhR) [41], hypoxia-inducible factor 1-alpha (Hif-1*α*) [42], as well as Egr2, Gata3, and Batf [30], play critical roles in *IL-10* gene regulation and IL-10 production. The integration of these regulatory pathways and transcription factors ultimately determines IL-10 levels and activity. However, disentangling their specific roles in IL-10 production, immune cell differentiation, or both, remains a challenge. For instance, using a novel computational single-cell morphology approach, macrophages have recently been classified into six phenotypes based on their intracellular IL-10 content. This innovative method provides a versatile framework for both cellular phenotyping and estimating IL-10 levels in IL-10-producing cells, contributing to a deeper understanding of cytokine biology [43].

## 2. Search Strategy

A systematic literature search was conducted using the electronic databases ‘PubMed’, ‘Scopus’, and ‘Web of Science’ in December 2024 to identify relevant studies on IL-10 immunotherapy and cancer. The search terms included ‘interleukin-10’ or ‘IL-10’, ‘Pegilodecakin’ with ‘cancer’, ‘tumour’, ‘neoplasia’, ‘immunotherapy’, ‘biological therapy’, ‘cell proliferation’, ‘apoptosis’, ‘metastasis’, ‘epithelial–mesenchymal transition (EMT)’, ‘tumour microenvironment (TME)’, ‘tumor-associated macrophages (TAMs)’, ‘tumor infiltrated immune cells’, and ‘cancer immunology’. These terms were used in various combinations to retrieve studies published from 2000 to the present. The reference lists of all selected articles were further reviewed to identify additional studies relevant to IL-10 in cancer treatment.

## 3. IL-10: A Historical Journey as a Tumor-Suppressor Cytokine

Cancer immunotherapy, including cytokine-based therapies, has revolutionized the landscape of cancer treatment over the past few decades. Among the most impactful cytokines, including IFN-α, IFN-γ, IL-2, IL-15, IL-7, IL-10, IL-12, and IL-21, extensive evaluations in preclinical and clinical trials have demonstrated significant cytotoxic T lymphocyte (CTL) responses and antitumor efficacy against various malignancies [44,45,46]. Notably, IFN-α and IL-2 have received FDA approval for cancer treatment, while others remain under investigation in preclinical models and ongoing clinical trials [45,46].

The protective role of IL-10 in preventing endogenous tumor development was initially identified in IL-10-deficient (IL-10^−/−^) mice, which spontaneously developed colon carcinoma [47]. The deficiency of IL-10R has also been linked to a predisposition to lymphoma, as patients with impaired IL-10 signaling often develop lymphomas within the first decade of life [48]. Experimental models have further demonstrated IL-10’s antitumor properties. When melanoma cells engineered to overexpress IL-10 were implanted into immunocompetent hosts, tumorigenicity was diminished, and tumor eradication was directly correlated with IL-10 secretion levels [49]. Similarly, in transgenic mice overexpressing IL-10 in antigen-presenting cells (IL-10TG), syngeneic tumors were rejected within weeks, a process critically dependent on CD8^+^ T cells [50]. The role of IL-10 in modulating tumor susceptibility was further demonstrated in squamous skin tumor models. IL-10^−/−^ mice exhibited earlier onset and greater tumor incidence compared to controls, whereas IL-10TG mice were protected from tumor formation [51]. Moreover, systemic administration of pegylated IL-10 (PEG-IL-10), a long-lived form of recombinant IL-10, succeeded to induce CD8^+^ T-cell–mediated tumor rejection across different murine tumor models, such as transplanted models of PDV6 squamous cell carcinoma, breast carcinoma, and melanoma, as well as chemically induced skin tumors [51,52,53].

As tumor suppressor, IL-10 exerts multiple mechanisms and pathways to eradicate tumor cells and enhance antitumor immune surveillance. It directly activates and expands tumor-resident and intratumoral infiltrating CD8^+^ T cells while augmenting their tumor-specific cytotoxic activity. Additionally, IL-10 promotes the intratumoral expression of key effector molecules, including IFN-γ, granzymes, and FasL, which are critical for tumor cell elimination [51,52,53,54]. Furthermore, IL-10 has been shown to inhibit CD8^+^ T cell apoptosis, thereby sustaining their antitumor functionality [55]. Beyond its direct effects on cytotoxic lymphocytes, IL-10 also modulates the TME by suppressing the synthesis of IL-23, downregulating IL-12p40 expression, and inhibiting TGF-β production [56,57]. The latter effect is mediated by limiting macrophage polarization and stromal cell activation, both of which are key drivers of TGF-β generation within the TME [23,24].

Translational research has explored IL-10-based therapies, including clinical trials evaluating PEG-IL-10 as a monotherapy or in combination with immune checkpoint inhibitors (ICIs) [58]. Additionally, emerging bioengineering strategies aim to enhance IL-10’s therapeutic potential by improving its stability, half-life, pharmacokinetics, and targeted-delivery and specificity to tumor sites, showing promising results in the preclinical studies. In this context, this review specifically underscores the progress and challenges in IL-10-directed cancer immunotherapies, highlighting recent advancements in the field.

## 4. Preclinical Strategies for Bioengineered IL-10 and Innovative Delivery Systems in Cancer Therapy

Various technologies have been employed to enhance the pharmacokinetic profile of IL-10, extending its half-life and improving its delivery and intratumoral accumulation. Pegylated proteins, fusion proteins, nanoparticles, vector delivery systems, and genetically modified IL-10 are revolutionizing cancer research landscape. Pegylation, which is the attachment of polyethylene glycol (PEG) to proteins, enhances protein stability, extends circulation time, and reduces immunogenicity, making it a powerful tool for cancer treatments [59]. Fusion proteins, which combine functional components of different proteins, enable the precise targeting of cancer cells while minimizing adverse effects [60]. Nanoparticles provide unique benefits, including targeted drug delivery and controlled release, which enhance therapeutic efficacy and reduce harm to healthy tissues [61]. Viral and non-viral vector delivery systems have enabled the localized and sustained delivery of therapeutic genes like IL-10 to tumor sites, enhancing its anti-inflammatory and tumor-modulating effects [62]. In our recent advancements, we generated engineered IL-10 proteins with improved stability and biological function, representing a critical step toward incorporating IL-10 into innovative delivery system platforms [63]. This work establishes a foundation for using IL-10 as a building block in cancer immunotherapy, improving its anti-inflammatory and anti-tumorigenic properties potential. Together, these strategies hold promise for developing targeted and effective cancer therapies.

The following section discusses preclinical findings on the most promising bioengineer approaches (Table 1) that have been developed to maximize the anticancer therapeutic potential of IL-10 by improving its tumor specificity and increasing its intratumoral concentrations [64]. These strategies include the development of IL-10-based bispecific fusion proteins, engineered IL-10 variants with increased affinity for IL-10Rβ, incorporation of IL-10 into oncolytic viruses, and conjugation of IL-10 to nanoparticles, excluding pegylated IL-10, as it has already been evaluated in multiple clinical trials, which are addressed later.

**Table 1 cancers-17-01012-t001:** Preclinical advances for IL-10–based strategies in cancer immunotherapy.

Strategy	Design and Therapeutic Agent(s)		Key Findings	Ref.
IL-10/Fc Fusion protein	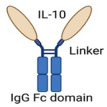	Fusion of IL-10 with IgG Fc domain	Prolonged IL-10’s half-life and well-tolerated. Potent antitumor activity, primarily mediated by reinvigorating and expanding exhausted CD8^+^ TILs. Synergized with ICIs or CAR T-cell therapy.	[65]
Bispecific CmAb(IL-10)_2_ Fusion Protein	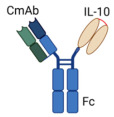	Fusion of Anti-EGFR antibody (Cetuximab “CmAb”) with IL-10 dimer to engage EGFR and IL-10R in EGFR^+^ tumors	Prolonged IL-10’s half-life and well-tolerated. Targeted EGFR-positive tumors. Improved tumor-specific cytotoxic activity of CD8^+^ T cells. Hindered DC-mediated apoptosis of CD8^+^ TILs. Overcame resistance to ICIs therapy.	[66]
BF10 Fusion Protein	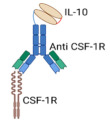	Bispecific Anti-CSF-IR/IL-10 Fusion Protein, combining IL-10 with anti-CSF-1R- antibody	Prolonged IL-10’s half-life and well-tolerated. Targeted TAMs-enriched tumors. Simultaneously delivered IL-10 selectively to TME while inhibiting CSF-1R signaling. Depleted TAMs and reprogrammed CD8^+^ TILs.	[67]
IL-10 variants for IL-10Rβ	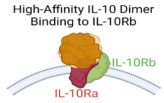	Bioengineered IL-10 variant with increased binding affinity for IL-10Rβ	Potent immunomodulatory effects at low doses. Enhanced cytotoxic activity of CAR T cells against acute myeloid leukemia cells.	[68]
Oncolytic Viruses (OVs) Based Strategy	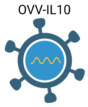	An oncolytic vaccina virus (OVV) armed with armed with IL-10 gene (OVV-IL10)	In pancreatic cancer models, OVV-IL-10 showed Sustained tumor suppression. Reduced risk of disease recurrence.	[69]
Nanoparticles & Scaffolds Based Systems	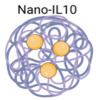	Conjugation of IL-10 into nanoparticles or scaffolds Delivery systems	Prolonged IL-10’s half-life. Well tolerated. Enhanced storage stability and antitumor activity.	[70,71]

Abbreviations: CAR T cells: Chimeric antigen receptor T cells, CSF-1R; colony-stimulating factor-1 receptor, EGFR; epidermal growth factor receptor, ICIs: immune checkpoint inhibitors, OVV; oncolytic vaccinia virus, TILs; tumor-infiltrating lymphocytes, TAMs; tumor-associated macrophages, TMEs; tumor microenvironments. Created in BioRender. Minshawi, F. (2025) [63] (https://BioRender.com/o45w353, accessed on 12 November 2024).

### 4.1. Fusion Proteins and Immunocytokine-Based Strategies

The fusion protein technology employed in the immunocytokine (ICK) approach presents a promising strategy to address the pharmacological limitations, suboptimal pharmacokinetics, and AEs associated with cytokine-based cancer therapies. These fusion proteins are engineered to enhance cytokine delivery specifically to the TME, thereby improving therapeutic efficacy [72,73]. Although multiple ICK strategies have been devised to stimulate antitumor immune responses [74], we herein focused on the most significant ICKs that specifically explore IL-10-based therapies in the preclinical cancer setting.

#### 4.1.1. Il-10/Fc Fusion Protein Approach

Genetic fusion of IL-10 with the fragment crystallizable (Fc) domain of the IgG antibody has led to the development of a therapeutic agent, termed the IL-10/Fc fusion protein. This new approach of IL-10-directed cancer therapy demonstrated low toxicity while exhibiting potent antitumor activity, primarily by reinvigorating and expanding terminally exhausted CD8^+^ tumor-infiltrating lymphocytes (TILs) [65]. Most importantly, these reinvigorated and expanded cytotoxic CD8^+^ TILs are crucial for tumor cell elimination. They are typically unresponsive to ICI therapy, representing a significant barrier in cancer immunotherapy [65].

Exhausted CD8^+^ TILs can be categorized into two subsets with distinct functional roles. Progenitor exhausted TILs (TCF-1+TIM-3−) possess proliferative potential and can differentiate into terminally exhausted TILs (TCF-1−TIM-3+), which directly contribute to tumor killing due to their superior cytotoxic activity [75,76,77]. While progenitor TILs respond to ICI therapies, such as anti-PD-1 blockade and mediate tumor suppression; terminally exhausted TILs lack proliferative capacity and remain largely unresponsive to ICIs and other immunotherapies [77,78]. Consequently, targeting the terminally exhausted CD8^+^ TILs subset within TME remains a significant challenge but offers immense therapeutic potential. Interestingly, combination therapy involving IL-10/Fc fusion protein and either ICI or chimeric antigen receptor (CAR) T-cell therapy has effectively reinvigorated terminally exhausted CD8^+^ TILs and demonstrated substantial tumor regression in mice with syngeneic models of melanoma, ovarian, and colorectal carcinoma [65]. These therapies were well-tolerated, with no significant weight loss or adverse toxic effects observed in the treated mice [65]. Moreover, the IL-10/Fc fusion protein has been shown to effectively shift and reprogram T cell metabolism via its mitochondrial pyruvate carrier system [65], and this metabolic shift is critical for reactivating terminally exhausted CD8^+^ TILs, enabling their functional recovery and antitumor activity [79].

#### 4.1.2. Bispecific CmAb-(IL-10)_2_ Fusion Protein: Targeting EGFR^+^ Tumors

To enhance the targeted delivery of IL-10 within the TME and overcome the limitations of its short half-life, Qiao et al. developed a bispecific fusion protein, designed as CmAb-(IL-10)_2_ [66]. This construction aimed to simultaneously engage an oncogenic receptor and the IL-10R. For proof-of-concept studies, the researchers utilized Cetuximab (CmAb), an FDA-approved monoclonal antibody that selectively blocks ligand binding and intracellular signaling of epidermal growth factor receptor (EGFR) [80]. Notably, EGFR, a type I receptor tyrosine kinase, and its ligands play a pivotal role in regulating multiple cellular pathways involved in cancer cell proliferation, survival, and metastasis, and in tumor angiogenesis [81,82]. Moreover, intratumoral excessive *EREG* mRNA expression is significantly linked with short OS in several human cancers [82]. Thus, the dual functionality of CmAb-(IL-10)_2_ is based on Cetuximab in one arm to bind EGFR^+^ tumor cells and an IL-10 dimer in the other arm to bind and activate IL-10R-mediated signaling pathways [66]. In murine tumor models of transplanted melanoma (B16-cEGFR), colorectal carcinoma (MC38-cEGFR), lung cancer (TC1-cEGFR), and human epidermoid carcinoma (A431-cEGFR)—where „cEGFR” indicates that these cancer cells were engineered to express a chimeric EGFR—CmAb-(IL-10)_2_ effectively extended the half-life of IL-10 without inducing toxicity. Additionally, it efficiently targeted EGFR^+^ tumors and specifically inhibited dendritic cell (DC)-mediated apoptosis of CD8^+^ TILs [66]. Mechanistic studies revealed that the antitumor efficacy of CmAb-(IL-10)_2_ was highly dependent on the presence of intratumoral CD8^+^ T cells and required IL-10R signaling on DCs to prevent DC-mediated apoptosis of CD8^+^ TILs. Furthermore, CmAb-(IL-10)_2_ demonstrated the capacity to overcome resistance to ICI therapy. When administered either intratumorally or systemically, CmAb-(IL10)_2_ exhibited a synergistic antitumor effect with ICI therapy, significantly improving therapeutic outcomes. These findings emphasize CmAb-(IL-10)_2_ as a promising therapeutic agent with potential for clinical translation [66]. In a constant line, a Phase 1/II clinical trial (NCT05396339 (https://clinicaltrials.gov/study/NCT05396339 (accessed on 27 November 2024)) is in recruiting status to investigate the safety, tolerability, dose escalation and pharmacokinetic profile, and the preliminary antitumor activity of IAE0972 (EGFR/IL10) in patients with advanced malignant solid tumors.

#### 4.1.3. Bifunctional Anti-CSF-1R/IL-10 Fusion Protein: Targeting TAMs-Enriched Tumors

Tumor-associated macrophages (TAMs) play a critical role in promoting tumor progression and metastatic colonization through the release of various tumorigenic mediators and growth factors. TAMs are predominant immune cell population within the TME, tumor blood vessels, and stromal regions, and their abundance is linked to poor clinical outcomes and resistance to therapies [83,84]. In addition, TAMs contribute to the induction of exhaustion programs inCD8^+^ TILs, whereas TAM depletion has been shown to reinvigorate and restore the functional capacity of exhausted CD8^+^ TILs [85]. Therefore, TAM depletion or repolarization is considered a critical target in cancer immunotherapy, particularly for solid tumors [86]. In this regard, the colony-stimulating factor 1 receptor (CSF-1R) is crucial in managing TAMs and other cellular interactions within the TME [87,88]. CSF-1R overexpression has been observed in various neoplastic conditions, where it is associated with TAMs polarization, tumor immune escaping, and a poorer prognosis [88,89]. Coherently, the inhibition of CSF-IR signaling has been strongly suggested for hindering immune suppression and escape in tumors [90]. In line with this, Chang et al. generated a bifunctional protein, referred to as BF10, which combines IL-10 with a CSF-1R-blocking antibody [67]. This bispecific construct aims to deliver IL-10 selectively to TAM-enriched TMEs while simultaneously inhibiting CSF-1R signaling. As a therapeutic agent, BF10 demonstrated significant antitumor activity across various preclinical cancer models, with particular efficacy against head and neck cancers [67]. Notably, BF10 not only induced TAMs depletion but also triggered the proliferation, activation, and metabolic reprogramming of CD8^+^ TILs. Moreover, combining BF10 with ICI therapy further amplified its antitumor efficacy. Biodistribution studies of BF10 revealed its accumulation in tumors, tumor-draining lymph nodes, and the spleen, pointing out its systemic and local anti-tumor immunomodulatory effects. These findings underscore the effective engineering strategy for BF10 therapy and its potential for clinical therapeutic development [67].

### 4.2. Bioengineering of IL-10 Variants with Enhanced Affinity for IL-10Rβ

IL-10 exhibits a high binding affinity for its IL-10Rα receptor subunits and a much lower affinity for its IL-10Rβ subunits [91,92,93]. This differential binding affinity has been proposed to play a crucial role in IL-10’s signaling mechanism and may contribute to the limited in vivo clinical efficacy of wild-type IL-10. Consequently, enhancing IL-10’s binding affinity to IL-10Rβ could improve its therapeutic potential. To investigate this hypothesis, Gorby et al. engineered an IL-10 variant with enhanced affinity for IL-10Rβ [68]. This variant more efficiently recruited IL-10Rβ into active cell surface signaling complexes, leading to stronger activation of STAT1 and STAT3 signaling pathways and increased transcriptional activity in human CD8^+^ T cells. Notably, these effects were observed even at low ligand concentrations [68]. Moreover, in the presence of this engineered IL-10 variant, chimeric antigen receptor (CAR) T cells expanded and exhibited significantly greater cytotoxicity against acute myeloid leukemia cells compared to those cultured with wild-type IL-10 [68]. Supporting this strategy, recent cryo-electron microscopy structural analysis of the IL-10 receptor complex revealed that the low affinity of IL-10 for IL-10Rβ ensures tightly regulated signaling, occurring only when IL-10 is present at sufficient concentrations [94,95]. Alternatively, the development of IL-10Rβ-specific agonist variants with modified cell-type selectivity may facilitate complete receptor complex formation and enhance IL-10’s clinical efficacy [68,94,95].

### 4.3. Oncolytic Viruses and Nanoparticle-Based Carriers for Il-10 and Other Biologically Active Il-10 Isoforms

Cancer virotherapy using oncolytic viruses (OVs) has witnessed a resurgence due to its selective replication and tumor-specific lytic effects, sparing normal cells, and its clinically established safety and tolerability profile [96]. Furthermore, we and others have reported that the arming of OVs with genes of pharmacologically active cytokines has ushered in a new generation of OVs combined with cancer immunotherapy, significantly enhancing tumor cell eradication and inducing a robust antitumor immune response [97,98,99,100]. Specifically, Chard et al. investigated the antitumor potential of an oncolytic vaccinia virus armed with IL-10 gene (OVV-IL10) in various pancreatic cancer models [69]. OVV-IL10 proved to be more effective in enhancing anti-tumor efficacy while dampening antiviral immune response compared to its unarmed counterpart. Moreover, it markedly exhibited a sustained tumor suppression and reduced risk of disease recurrence, achieved through modulation of both innate and adaptive immune responses and stimulation of anti-tumor cytotoxic CD8^+^ T cells [69].

Another promising strategy involves nanoparticle-mediated IL-10 delivery, such as its conjugation with polyvinylpyrrolidone-coated silver-based nanoparticles [70]. This approach has demonstrated the ability to enhance storage stability while preserving the biological functionality of IL-10 [70]. Similarly, scaffold-based delivery systems, including microspheres and hydrogels, have been extensively studied for cytokine delivery. These systems can be injected or implanted directly at the diseased site, enabling controlled and sustained release of cytokines as the scaffold material gradually degrades [101,102]. This controlled release not only prolongs the half-lives of the loaded cytokines but also minimizes their systemic toxicity. Moreover, scaffold-based approaches can allow for the co-delivery of multiple therapeutic agents within a single matrix, and the simultaneous delivery of these agents ensures spatial and temporal coordination, thereby fostering synergistic therapeutic effects and improving treatment efficacy [71,103].

Recently, our research team developed a genetically modified IL-10 variant (stable IL-10) with enhanced stability and biological activity at significantly lower concentrations than natural IL-10 [63]. Furthermore, we are currently designing a targeted delivery system for this stable IL-10 isoform by conjugating it to tumor-specific antibodies (e.g., anti-EGFR) to improve intratumoral accumulation and therapeutic efficacy in preclinical models of colorectal carcinoma. This strategy builds upon our prior success in delivering stable IL-10 to colonic inflammatory sites using anti-VCAM antibodies (unpublished data). However, the safety profile of our genetically engineered IL-10, both with and without antibody conjugation, remains to be evaluated, along with its anticancer efficacy alone and in combination with conventional therapy.

Other approaches to improve the anticancer pharmacological profile of therapeutic IL-10 have also been proposed. In this context, the complement receptors C3aR and C5aR, which are highly expressed in tumor and immune cells, have recently been identified as immune checkpoint receptors with critical roles in tumor development and progression. Their aberrant intratumoral activation fosters chronic inflammation, angiogenesis, tumor cell proliferation, survival, and invasion [104,105,106,107]. Crucially, activation of these receptors suppresses IL-10 production and inhibits IL-10-dependent T-cell-mediated antitumor immunity. Conversely, blocking these signaling pathways reverses the transcriptional suppression of IL-10, stimulates IL-10 production by antitumor T cells, and enhances T-cell expansion and antitumor activity [108]. Accordingly, a combination approach involving IL-10 and C3aR/C5aR antagonists could serve as an effective therapeutic strategy for immune-resistant cancers, particularly when augmented with the addition of IL-2 [IL-10 + IL-2 + C3aR/C5aR signaling blockade] [109]. This proposed strategy is further supported by earlier findings that demonstrated the in vitro synergy between IL-10 and IL-2 in expanding and enhancing the activity of CD8^+^ T cells derived from lung cancer patients. The IL-10/IL-2 combination not only increased their IFN-γ expression but also upregulated genes associated with cytotoxic antitumor activity [108,109].

In summary, the intratumoral delivery methods and sustainability of therapeutic IL-10 (or its biologically active isoforms), specifically designed to interact with critical targets within the TME, are critical considerations to effectively mediate multimodal antitumor activity within tumor tissues while reducing systemic adverse effects and minimizing inhibitory impacts on non-target cell types.

## 5. Clinical Trials of Il-10 Directed in Cancer Immunotherapy

The multifaceted role of sustained IL-10 in simultaneously activating and expanding cytotoxic tumor-specific CD8^+^ T cells while suppressing tumor-associated inflammation has positioned it as a cytokine of significant clinical interest in cancer immunotherapy. To maximize IL-10’s therapeutic potential, various bioengineering strategies are being actively investigated, with PEGylation representing a key advancement in its clinical translation [110].

PEG, an FDA-approved polymer widely used to enhance the pharmacokinetics of therapeutic proteins, improves drug stability by extending circulation half-life, reducing renal clearance, and lowering systemic dosing requirements, thereby mitigating dose-limiting toxicities [74,111]. Specifically, PEGylation extends IL-10’s half-life from only a few hours [112,113] to approximately 24 h, ensuring sustained serum and intratumoral IL-10 concentrations while preserving its dual antitumor and immunoregulatory functions [114]. Within this framework, pegilodecakin (AM0010) has been the most extensively studied PEGylated IL-10 formulation in clinical cancer therapy (Table 2).

**Table 2 cancers-17-01012-t002:** Summary of published clinical trials on pegilodecakin (PEGylated IL-10) therapy.

Trial (NCT) and [Ref.]	Phase, Patients, and Design	Therapeutic Agent(s)	Key Outcomes	Interpretation
IVY (NCT02009449) [115]	Phase 1; patients with various advanced solid tumors	PEG alone	An acceptable safety and early antitumor activity	Warranting further evaluation
IVY (NCT02009449) [116]	Phase 1b; patients with various solid tumors, including patients with treatment-refractory NSCLC and RCC	PEG + Anti-PD1 therapy (PEMB or NIVO)	ORR: 43% (NSCLC) and 40% (RCC)	Favorable anti-tumor activity between PEG and Anti-PD1 therapy
IVY (NCT02009449) [117]	Phase 1b; mRCC patients	PEG + NIVO *vs.* PEG + Pazopanib	ORR: 43% (PEG/NIVO) *vs*. 33% (PEG/Pazopanib)	PEG/NIVO showed better activity and tolerable safety profile
Ovarian Cancer [118]	Phase II; patients with metastatic Ovarian cancer	PEG alone *vs.* PEG + platinum-taxane	PFS:2.4 months (PEG) *vs*. 5.2 months (co-therapy)	Warranting further evaluation
CYPRESS 1 and 2 (NCT03382899 and (NCT03382912) [119]	Phase II; mNSCLC patients divided into: CYPRESS 1 (PD-L1 TPS ≥ 50%) and CYPRESS 2 (PD-L1 TPS 0–49%)	PEG + PEMB (or NIVO) *vs.* PEMB (or NIVO) alone	CYPRESS 1: ORR (47% *vs.* 44%) and PFS (6.3 *vs*. 6.1 months) and CYPRESS 2: ORR (15% *vs*. 12%), PFS (1.9 *vs*. 1.9 months), and OS (6.7 *vs*. 10.7 months)	PEG + Anti-PD1 therapy (PEMB or NIVO) did not offer benefits over Anti-PD1 alone
SEQUOIA (NCT02923921) [120,121]	Phase III; gemcitabine-refractory PDAC patients	PEG + FOLFOX *vs.* FOLFOX alone	ORR: 4.6% *vs*. 5.6%; PFS: 2.1 *vs*. 2.1 months; and OS: 5.8 *vs*. 6.3 months	PEG did not improve FOLFOX efficacy in advanced gemcitabine-refractory PDAC patients

Abbreviations: AEs; Adverse events, FOLFOX; 5-fluorouracil, leucovorin calcium, oxaliplatin, mNSCLC; metastatic non-small cell lung cancer, mRCC; metastatic renal cell carcinoma; NIVO; nivolumab, ORR; Objective tumor response, OS; overall survival, PEG: pegilodecakin, PEMB; pembrolizumab, PDAC: pancreatic ductal adenocarcinoma; PD-L1 TPS; proportion score of PD-L1–positive tumor cells; PFS; progression-free survival.

### 5.1. Early-Phase Dose Escalation Study of Pegilodecakin Monotherapy

In Phase 1 of the first-in-human trial, IVY (NCT02009449 (https://clinicaltrials.gov/study/NCT02009449 (accessed on 26 November 2024)), a total of 51 patients with diverse advanced solid tumors who had failed all available prior therapies, including previous ICIs or IL-2 treatments, were enrolled and treated daily with subcutaneous self-injected pegilodecakin monotherapy at escalating doses ranging from 1 μg/kg to 40 μg/kg [115]. Objective tumor response (ORR) was assessed based on immune-related response criteria [122]. Overall, pegilodecakin monotherapy demonstrated encouraging antitumor responses, particularly in patients with uveal melanoma or advanced renal cell carcinoma (RCC). Delayed, prolonged, and mixed antitumor responses were observed in other enrolled patients, including durable disease stability in a patient with microsatellite-stable colorectal cancer [114,115]. Notably, the administered pegilodecakin led to a sustained and clinically meaningful elevation in serum IL-10 concentrations, ranging between 1 and 40 ng/mL. The most frequently observed adverse events (AEs) were mild to moderate (grade 1 or 2) and included injection-site reactions, fever, fatigue, anemia, and thrombocytopenia, with seven patients in the expansion cohort necessitating dose reductions. Grade 3 or 4 anemia, thrombocytopenia, transaminitis, and dermatologic reactions, were reported but were rare and resolved without complications [114,115]. Importantly, hypotension and long-lasting immune-related adverse events (irAEs) were not reported with pegilodecakin monotherapy. Furthermore, consistent with IL-10’s known anti-inflammatory properties, inflammatory irAEs such as hepatitis, colitis, or pneumonitis were also not observed. This contrasts with ICIs or IL-2 treatments, where ICIs frequently induce irAEs [123], and IL-2 is associated with dose- and duration-limiting severe hypotension [124]. Moreover, pharmacodynamic analyses revealed that pegilodecakin-treated patients achieved sustained IL-10 levels and exhibited significant upregulation of IFNγ and IL-18, accompanied by a 42% reduction in serum TGFβ levels [114,115]. These findings support the hypothesis that pharmacologically stabilized IL-10 enhances T-cell activation in cancer patients.

### 5.2. Evaluation of Pegilodecakin in Combination with Anti-PD-1 Therapy

In Phase 1b of the IVY trial (NCT02009449 (https://clinicaltrials.gov/study/NCT02009449 (accessed on 26 November 2024)), 111 patients were enrolled and received combination therapy with pegilodecakin plus ICIs (pembrolizumab or nivolumab anti-PD-1- monoclonal antibodies). The patient cohorts included those with advanced and treatment-refractory RCC (*n* = 38), melanoma (*n* = 37), non-small cell lung cancer (NSCLC, *n* = 34), bladder cancer (*n* = 1), and triple-negative breast cancer (*n* = 1). The achieved ORR was 43% in NSCLC, 40% in RCC, and 10% in melanoma [116]. Moreover, the combination of pegilodecakin and anti-PD-1 therapy resulted in a progression-free survival (PFS) of 9.4 months and an overall survival (OS) of 24.1 months in NSCLC patients, as well as a PFS of 16.7 months and 1-year OS of 93% in patients with RCC [116]. These findings contrast with those observed in earlier studies involving a similar population of RCC patients who were mono-treated with anti-PD-1 (nivolumab) and showed an ORR of 22%, a PFS of 4.2 months and < 80% 1-year OS [125]. Similarly, in patients with advanced NSCLC, who received anti-PD-1 (pembrolizumab) monotherapy, the ORR was 19.4%, with a PFS of 3.7 months and an OS of 12 months [126]. Moreover, in these earlier studies, the response to anti-PD-1 monotherapy was strongly correlated with high PD-L1 expression in tumor cells [126]. In contrast, responses to pegilodecakin–anti-PD-1 combination therapy in the Phase 1b IVY trial were observed even in patients with low PD-L1 expression [116]. The AEs profile across patients’ cohorts was aligned with the prior data reported for single-agent pegilodecakin or anti-PD-L1 therapies. Red blood cell hemophagocytosis and lymphohistiocytosis were identified in three patients (grade 1/2 in two patients, grade 4 in one patient), all of whom recovered uneventfully following resolution of these events [116].

Next, a comprehensive analysis was focused on all metastatic RCC (mRCC) patients in the above IVY trial (NCT02009449 (https://clinicaltrials.gov/study/NCT02009449 (accessed on 26 November 2024)) [117]. Specifically, the combination of pegilodecakin with anti-PD-1 therapy resulted in promising clinical outcomes and achieved an ORR of 43%, compared to 33% for the combination of pegilodecakin with the multi-tyrosine kinase inhibitor “pazopanib”. Additionally, while pegilodecakin monotherapy in these mRCC patients resulted in a PFS of 1.8 months and OS probabilities of 50% at one year and 17% at two years, its combination with anti-PD-1 significantly led to a PFS of 13.9 months and OS probabilities of 76% at one year and 61% at two years. Furthermore, the AEs associated with pegilodecakin, both as monotherapy and in combination with ICIs, were consistent with prior reports [117]. Since most patients in the IVY trial were heavily pretreated and had experienced disease progression prior to their enrollment, these findings may hold clinical significance, particularly in the context of mRCC therapy, emphasizing the need for further investigations [117]. In support, an interim report evaluated the clinical activity of pegilodecakin monotherapy in metastatic epithelial ovarian cancer reported that 44% of the treated patients achieved a disease control rate of 66.7% with PFS exceeding 3.5 months [114,118].

### 5.3. Evaluation of Pegilodecakin as an Add-On Therapy in CYPRESS and SEQUOIA Clinical Trials

Despite the encouraging outcomes reported above in the IVY study, pegilodecakin failed to demonstrate promising results in the subsequent CYPRESS 1 and 2 clinical trials (NCT03382899(https://clinicaltrials.gov/study/NCT03382912 (accessed on 23 October 2024)); NCT03382912 (https://clinicaltrials.gov/study/NCT03382899 (accessed on 23 October 2024)). These randomized Phase II studies investigated whether the addition of pegilodecakin to immune checkpoint inhibitors (ICIs; anti-PD-1 monoclonal antibodies: pembrolizumab or nivolumab) offered therapeutic benefits for patients with metastatic NSCLC compared to ICIs alone [119]. Specifically, CYPRESS 1 enrolled patients with high PD-L1 expression (tumor proportion score [TPS] ≥ 50%), while CYPRESS 2 included patients with low or negative PD-L1 expression (TPS 0–49%) [119]. In CYPRESS 1, patients received either pembrolizumab monotherapy (*n* = 50) or a combination of pegilodecakin and pembrolizumab (*n* = 51), while in CYPRESS 2, treatments included nivolumab alone (*n* = 25) or combined with pegilodecakin (*n* = 27). Pegilodecakin was self-administered subcutaneously by the patents at fixed doses of 0.4 mg (for patients ≤ 80 kg) or 0.8 mg (for patients > 80 kg) on days 1–5 and 8–12, with rest periods on days 6–7 and 13–14, as guided by the Phase I IVY study findings. Overall, the pegilodecakin + anti-PD-1 combination therapy did not significantly improve ORR, PFS, or OS compared to anti-PD-1 monotherapy. For CYPRESS 1 (pegilodecakin + pembrolizumab vs. pembrolizumab alone), the results were ORR (47% vs. 44%), PFS (6.3 vs. 6.1 months), and OS (16.3 months vs. not reached). For CYPRESS 2 (pegilodecakin + nivolumab vs. nivolumab alone), the results were ORR (15% vs. 12%), PFS (1.9 vs. 1.9 months), and OS (6.7 vs. 10.7 months). Furthermore, treatment discontinuation rates due to severe hematologic AEs were doubled in the combination treatment arms of both studies [119].

In a constant line, the SEQUOIA study (NCT02923921(https://clinicaltrials.gov/study/NCT02923921 (accessed on 12 November 2024)), a global randomized Phase III trial, assessed the efficacy and safety of adding pegilodecakin to the chemotherapy regimen of folinic acid, fluorouracil, and oxaliplatin (FOLFOX) as a second-line treatment for gemcitabine-refractory metastatic pancreatic ductal adenocarcinoma (PDAC). Patients were randomized to receive pegilodecakin + FOLFOX (*n* = 283) or FOLFOX alone (*n* = 284). Alongside the standardized FOLFOX regimen, pegilodecakin was self-administered subcutaneously at fixed doses as described above. Although the pegilodecakin + FOLFOX combination initially demonstrated enhanced antitumor activity compared to FOLFOX alone—indicated by immunostimulatory signals of the IL-10R pathway, including increased levels of IFN-γ and granzyme B and decreased TGF-β levels—its overall results showed no improvement in OS (5.8 vs. 6.3 months), PFS (2.1 vs. 2.1 months), and ORR (4.6% vs. 5.6%) [120,121]. Moreover, the incidence of grade ≥3 hematological AEs, particularly anemia, thrombocytopenia, and neutropenia, was notably higher in the pegilodecakin + FOLFOX arm, necessitating greater use of red blood cell transfusions and erythropoiesis-stimulating agents (5.8% vs. 2.0%). These findings underscore the need for further translational research to identify patients likely to benefit from pegilodecakin-chemotherapy combination in treatment-refractory advanced tumors [120,121].

Taken together, while the PEGylation process of recombinant human IL-10 (pegilodecakin) has notably enhanced its stability, serum concentration, and half-life, all while maintaining its biological activity, its overall benefits in the CYPRESS and SEQUOIA clinical trials has exhibited significant limitations [119,120,121]. The heterogeneity in cancer types, patient populations, and therapeutic combination types and regimens might complicate direct comparisons among these trials [58]. Importantly, although PEGylation is clinically approved for IFNα2a and IL-2 therapy and is currently under evaluation for additional immune-cytokine therapies such as IL-10 and IL-15, it is often associated with unwanted dose-dependent effects on bioactivity, primarily due to its amphiphilic nature [74]. In support, PEG-IL10-induced dose-dependent toxicity has been reported in both preclinical and clinical studies [53]. To overcome these challenges, further investigations, including the development of alternative and reprogrammed strategies optimizing IL-10 delivery and intratumoral localization, are warranted. These efforts are essential to fully harness and realize the mono- or combination-therapeutic potential of IL-10-directed therapy across a range of malignancies.

## 6. Conclusion

While IL-10 has historically been associated with tumor progression, accumulating evidence underscores its ability to enhance antitumor immunity by reinvigorating exhausted CD8^+^ TILs and activating their cytotoxic function; inducing IFN-γ, IL-18, granzymes, and FasL expression; reducing tumor-associated inflammation and TGF-β levels; and reprogramming TAMs. These attributes have spurred interest in IL-10-based therapies, both as monotherapy and in combination with ICIs.

Advancements in IL-10 bioengineering are driving its clinical translation. PEGylated IL-10 (pegilodecakin) demonstrated potential in early clinical trials, particularly in patients with RCC [114,116,117]. However, later-stage clinical trials, including CYPRESS and SEQUOIA, revealed challenges such as dose-limiting toxicities and inconsistent efficacy [119,120,121]. These findings underscore the necessity of refining IL-10-based therapeutics through improved delivery mechanisms, optimized dosing strategies, and precise patient selection criteria.

Preclinical advancements offer promising strategies to overcome these limitations [64]. Engineered IL-10 variants with enhanced IL-10Rβ affinity have demonstrated superior stability and antitumor activity in murine models. Additionally, IL-10 fusion proteins, such as IL-10/Fc constructs, have shown potential in prolonging IL-10’s half-life, while minimizing systemic exposure, and in reinvigorating exhausted CD8^+^ TILs, thereby enhancing tumor-specific immune responses [65]. The development of bispecific fusion proteins, such as CmAb-(IL-10)_2_ [66] and BF10 [67], further supports the rationale for IL-10-based combination therapies and exhibiting remarkable synergy with ICIs in preclinical studies. Emerging delivery platforms, including nanoparticle-based and scaffold-mediated IL-10 formulations, provide additional avenues to improve IL-10’s therapeutic efficacy. Collectively, such approaches hold promise for overcoming the pharmacokinetic challenges associated with IL-10 therapy, enabling controlled and sustained intratumoral IL-10 release, ensuring localized immune activation with reduced toxicity, and warranting further investigation in translational and clinical settings. Looking ahead, the integration of IL-10 into precision oncology frameworks requires a deeper understanding of its mechanistic role in shaping the TME. Future research should focus on optimizing IL-10-based combination regimens, leveraging predictive biomarkers for patient stratification, and refining delivery strategies to maximize therapeutic efficacy. Additionally, elucidating the interplay between IL-10 and other immunoregulatory pathways will provide valuable insights into its broader applicability across various cancer types.

In conclusion, IL-10 represents a compelling avenue in cancer immunotherapy, with the potential to reshape treatment paradigms. Enhancing tumor specificity while minimizing systemic toxicity remains a critical objective for future clinical applications. Continued innovation in cytokine engineering, targeted delivery, and combination strategies will be instrumental in overcoming existing limitations and unlocking IL-10’s full therapeutic potential. As research progresses, harnessing IL-10’s immunomodulatory capabilities may pave the way for more effective, durable, and personalized cancer treatments.

## Data Availability

The data that support the findings of this study are available within the article.

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
