# Peer review of "IL-10-Directed Cancer Immunotherapy: Preclinical Advances, Clinical Insights, and Future Perspectives"

_cancers, 2025, doi:10.3390/cancers17061012_

Round 1
Reviewer 1 Report
Comments and Suggestions for Authors
This is an interesting review that presents the current state of the art regarding the use of IL-10 in cancer treatment. It is well-written, thoroughly discussed, and well-illustrated. However, despite the encouraging preclinical results, I am concerned that, in an increasingly competitive landscape where testing new molecules in combination with clinical standards is difficult, the challenges posed by negative results in Phase 2 and 3 trials make it highly complicated to further evaluate IL-10 as a viable option. Even with potential R&D improvements as discussed by the authors, this is something to consider carefully.
Below are a few suggestions for improvement:
1-In both the introduction and discussion, it would be valuable to include a bit more on the historical development of cytokine-based strategies in immunotherapy (e.g., IL-2, IL-15, IL-7, etc.). It would also be useful to position IL-10 in relation to these other cytokines and explain how it compares to other similar therapeutic products.
2-It would be helpful to emphasize any new studies currently enrolling patients or those planned for the near future.
3-In the introduction, Authors claim "IL-10 is produced by a diverse array of cells." Could you provide more details on the levels of IL-10 produced by different cell types? Is there variability in production or secretion between these cells? Further development of this point would provide a clearer understanding of its biological complexity
4-On which cells are the IL-10 receptors expressed? It would be helpful to clarify this to better understand the targeting and potential therapeutic implications of IL-10.
5-The introduction seems to focus primarily on IL-10's anti-tumor effects. Perhaps a dedicated paragraph would be worthy
6-Throughout the manuscript, please provide more precise information on the specific mouse and tumor models used in the different preclical research studies. Exemple when mentioning "L101 across multiple murine tumor models [31,32]," similarly in lines 190–200 and 214–216, details on the model systems would enhance the reader's understanding of the experimental context and readability
7-There is a minor typo in the text: "theIL-10 gene" should be corrected to "the IL-10 gene."
8-In line 302, authors mention the design of a targeted delivery system for the stable IL-10 isoform using tumor-specific antibodies (e.g., anti-EGFR). It would be helpful to clarify whether this system is designed to target specific tumor types. More precise information about the intended targets and the scope of this development would be beneficial.
9-Table 2 is somewhat difficult to read in the PDF document. It may be worth considering reformatting or improving the presentation for clearer readability
10-When discussing clinical trials, please provide more detailed information on the treatment modalities for IL-10 derivatives. Specifically, the number of injections, frequency, dose ranges, when available etc. For example, trials involving PEGylated forms of IL-10 could be explored in more detail. It should be possible to collect and present additional data regarding these aspects, which would improve the practical understanding of IL-10's therapeutic potential or failure
Author Response
Reviewer #1 (Comments and Suggestions for Authors):
This is an interesting review that presents the current state of the art regarding the use of IL-10 in cancer treatment. It is well-written, thoroughly discussed, and well-illustrated. However, despite the encouraging preclinical results, I am concerned that, in an increasingly competitive landscape where testing new molecules in combination with clinical standards is difficult, the challenges posed by negative results in Phase 2 and 3 trials make it highly complicated to further evaluate IL-10 as a viable option. Even with potential R&D improvements as discussed by the authors, this is something to consider carefully.
Below are a few suggestions for improvement:
Reviewer comment 1: In both the introduction and discussion, it would be valuable to include a bit more on the historical development of cytokine-based strategies in immunotherapy (e.g., IL-2, IL-15, IL-7, etc.). It would also be useful to position IL-10 in relation to these other cytokines and explain how it compares to other similar therapeutic products
Response: We sincerely appreciate the reviewer’s thoughtful feedback and valuable suggestion. Indeed, the historical development and the current clinical situation of cytokines-based cancer immunotherapies, highlighting and comparing their similarities and distinctions, is a crucial topic that warrants further exploration (e.g., "Cytokine-Based Cancer Immunotherapy: Where Are We Now and Where Are We Headed?").
In response to this suggestion, we have expanded the revised manuscript by including these three additions to enhance the reader’s understanding:
- Page 3, lines 129–135: “Cancer immunotherapy, including cytokine-based therapies, has revolutionized the landscape of cancer treatment over the past few decades. Among the most impactful cytokines, including IFN-α, IFNγ, IL-2, IL-15, IL-7, IL-10, IL-12, and IL-21, extensive evaluations in preclinical and clinical trials have demonstrated significant cytotoxic T lymphocyte (CTL) responses and antitumor efficacy against various malignancies [44-46]. Notably, IFN-α and IL-2 have received FDA approval for cancer treatment, while others remain under investigation in preclinical models and ongoing clinical trials [45,46].”
- Page 9, lines 361–368: “Accordingly, a combination approach involving IL-10 and C3aR/C5aR antagonists could serve as an effective therapeutic strategy for immune-resistant cancers, particularly when augmented with the addition of IL-2 [IL-10 + IL-2 + C3aR/C5aR signaling blockade] [109]. This proposed strategy is further supported by earlier findings that demonstrated the in vitro synergy between IL-10 and IL-2 in expanding and enhancing the activity of CD8+ T cells derived from lung cancer patients. The IL-10/IL-2 combination not only increased their IFN-γ expression but also upregulated genes associated with cytotoxic antitumor activity [108,109].”
- Page 12, lines 361–368: “Importantly, although PEGylation is clinically approved for IFNα2a and IL-2 therapy and is currently under evaluation for additional immune-cytokine therapies such as IL-10 and IL-15, it is often associated with unwanted dose-dependent effects on bioactivity, primarily due to its amphiphilic nature [67].”
Reviewer comment 2: It would be helpful to emphasize any new studies currently enrolling patients or those planned for the near future.
Response: We sincerely than reviewer for this suggestion. We have now incorporated a subsection highlighting ongoing studies aiming to utilize complement system components to mediate intratumoral direction of IL-10-based cancer therapy (Page 2, lines 352–373): “Other approaches to improve the anti-cancer pharmacological profile of therapeutic IL-10 have also been proposed. In this context, the complement receptors C3aR and C5aR, which are highly expressed in tumor and immune cells, have recently been identified as immune checkpoint receptors with critical roles in tumor development and progression. Their aberrant intratumoral activation fosters chronic inflammation, angiogenesis, tumor cell proliferation, survival, and invasion [104-107]. Crucially, activation of these receptors suppresses IL-10 production and inhibits IL-10-dependent T-cell-mediated antitumor immunity. Conversely, blocking these signaling pathways reverses the transcriptional suppression of IL-10, stimulates IL-10 production by antitumor T cells, and enhances T-cell expansion and antitumor activity [108]. Accordingly, a combination approach involving IL-10 and C3aR/C5aR antagonists could serve as an effective therapeutic strategy for immune-resistant cancers, particularly when augmented with the addition of IL-2 [IL-10 + IL-2 + C3aR/C5aR signaling blockade] [109]. This proposed strategy is further supported by earlier findings that demonstrated the in vitro synergy between IL-10 and IL-2 in expanding and enhancing the activity of CD8+ T cells derived from lung cancer patients. The IL-10/IL-2 combination not only increased their IFN-γ expression but also upregulated genes associated with cytotoxic antitumor activity [108,109].
In summary, the intratumoral delivery methods and sustainability of therapeutic IL-10 (or its biologically active isoforms), specifically designed to interact with critical targets within the TME, are critical considerations to effectively mediate multimodal antitumor activity within tumor tissues while reducing systemic adverse effects and minimizing inhibitory impacts on non-target cell types.”
Reviewer comment 3: In the introduction, Authors claim "IL-10 is produced by a diverse array of cells." Could you provide more details on the levels of IL-10 produced by different cell types? Is there variability in production or secretion between these cells? Further development of this point would provide a clearer understanding of its biological complexity
Response: We sincerely thank the reviewer for this thoughtful comment. Due to the inherent complexity and variability of this topic, precise data on the production levels of IL-10 across different cell types remain limited. It is evident that variability exists not only between distinct cell types but also among different phenotypes within the same cell type. For example, using a novel computational single-cell morphology approach, macrophages have recently been classified into six phenotypes based on their intracellular IL-10 content [Ref. 43 in the revised manuscript]. Furthermore, IL-10 production is regulated by several factors, including external stimuli, cellular differentiation states, and the intricate interplay of regulatory signalling pathways and transcription factors, all of which collectively influence the variability of IL-10 expression in immune cells. Disentangling the specific contributions of these regulatory factors to IL-10 production versus their roles in immune cell differentiation also presents a significant challenge.
In response to this valuable comment, we have expanded our introduction section to a detailed overview of the diverse factors influencing IL-10 secretion by its heterogeneous cellular subsets (Page 3, lines 93–116): “Given its diverse sources, multifaceted functions, and the multiple variables that can modulate its signaling activity, targeting IL-10 for cancer immunotherapy presents a formidable challenge [27]. Notably, IL-10 expression by its heterogeneous cellular subsets can be modulated at multiple levels, including chromatin remodeling, transcriptional control, post-transcriptional processing, and cellular differentiation states. While some regulatory mechanisms are conserved across cell types, others exhibit cell type- or stimulus-specific variability [28-30]. For instance, innate immune cells rely on pathogen-derived products, such as those recognized by pattern recognition receptors (PRRs), including Toll-like receptors (TLRs), to induce IL-10 production [31]. Modulation by autocrine or paracrine cytokines further influences IL-10 production in these innate immune cells [28]. In contrast, naïve CD4+ T cells require differentiation into subsets before acquiring the ability to produce IL-10 [32]. Furthermore, several signaling pathways influence IL-10 production, including MAPK/ERK1/2 [33,34], PI3K-AKT [35], NF-κB [36], and mTOR pathways [37]. Similarly, transcription factors such as c-Maf [38], Jun family members (JunB, c-Jun) [39], Foxp3 [40], aryl hydrocarbon receptor (AhR) [41], hypoxia-inducible factor 1-alpha (Hif-1α) [42], as well as Egr2, Gata3, and Batf [30], play critical roles in IL-10 gene regulation and IL-10 production. The integration of these regulatory pathways and transcription factors ultimately determines IL-10 levels and activity. However, disentangling their specific roles in IL-10 production, immune cell differentiation, or both, remains a challenge. For instance, using a novel computational single-cell morphology approach, macrophages have recently been classified into six phenotypes based on their intracellular IL-10 content. This innovative method provides a versatile framework for both cellular phenotyping and estimating IL-10 levels in IL-10-producing cells, contributing to a deeper understanding of cytokine biology [43].”
Reviewer comment 4: On which cells are the IL-10 receptors expressed? It would be helpful to clarify this to better understand the targeting and potential therapeutic implications of IL-10.
Response: We sincerely appreciate the reviewer’s insightful comment. To address this, we have expanded the revised manuscript by including the following information (Page 3, lines 77–83): “IL-10Rs are expressed on various immune-associated hematopoietic cells as well as non-immune cells, such as epithelial cells, endothelial cells, and fibroblasts. However, their expression patterns vary depending on the cell type and specific biological context, and this diversity enables IL-10 to exert a wide range of signaling activities [13, 14]. Moreover, the expression of IL-10Rs, particularly IL-10R1, is tightly regulated and can be upregulated in response to inflammatory stimuli or activation of specific signaling pathways [13,14].“
Reviewer comment 5: The introduction seems to focus primarily on IL-10's anti-tumor effects. Perhaps a dedicated paragraph would be worthy
Response: We sincerely appreciate the reviewer’s insightful comment. In response, we have incorporated a dedicated paragraph in the revised manuscript (Page 4, Line 128) titled "IL-10: A Historical Journey as a Tumor-Suppressor Cytokine." We believe this addition provides a more structured overview of IL-10’s anti-tumor properties and historical perspective, and in turn, it improves the introduction's balance and strengthens the contextual foundation of our review.
Reviewer comment 6: Throughout the manuscript, please provide more precise information on the specific mouse and tumor models used in the different preclical research studies. Exemple when mentioning "L101 across multiple murine tumor models [31,32]," similarly in lines 190–200 and 214–216, details on the model systems would enhance the reader's understanding of the experimental context and readability
Response: We sincerely appreciate the reviewer’s insightful suggestion. In response, we have incorporated more precise details on the specific mouse and tumor models used in the referenced preclinical research studies to enhance the clarity and contextual understanding of the experimental findings. The revisions are as follows:
- (Page 4, Lines 151, 152): “murine tumor models, such as transplanted models of PDV6 squamous cell carcinoma, breast carcinoma, and melanoma, as well as chemically induced skin tumors [51-53].”
- (Page 7, Line 235): “mice with syngeneic models of melanoma, ovarian, and colorectal carcinoma [68].”
- (Page 7, Lines 253-259): “In murine tumor models of transplanted melanoma (B16-cEGFR), colorectal carcinoma (MC38-cEGFR), lung cancer (TC1-cEGFR), and human epidermoid carcinoma (A431-cEGFR)—where "cEGFR" indicates that these cancer cells were engineered to express a chimeric EGFR—CmAb-(IL-10)â‚‚ effectively extended the half-life of IL-10 without inducing toxicity. Additionally, it efficiently targeted EGFR+ tumors and specifically inhibited dendritic cell (DC)-mediated apoptosis of CD8+TILs [74].”
Reviewer comment 7: There is a minor typo in the text: "theIL-10 gene" should be corrected to "the IL-10 gene."
Response: We sincerely appreciate the reviewer’s attention to detail. In response, we have corrected this typographical error in the revised manuscript (Page 2, Line 66) to "the IL-10 gene" Thank you for your careful review.
Reviewer comment 8: In line 302, authors mention the design of a targeted delivery system for the stable IL-10 isoform using tumor-specific antibodies (e.g., anti-EGFR). It would be helpful to clarify whether this system is designed to target specific tumor types. More precise information about the intended targets and the scope of this development would be beneficial.
Response: We sincerely appreciate the reviewer’s insightful attention to detail. In response, we have clarified the intended tumor targets in the revised manuscript (Page 9, Lines 346–351), specifying that this targeted delivery system using tumor-specific antibodies (e.g., anti-EGFR) is being evaluated in preclinical models of colorectal carcinoma. We appreciate the reviewer’s valuable feedback.
Reviewer comment 9: Table 2 is somewhat difficult to read in the PDF document. It may be worth considering reformatting or improving the presentation for clearer readability
Response: We sincerely thank the reviewer for their thoughtful attention to detail. In response, we have reformatted and improved the presentation of Table 2 in the revised manuscript (Page 10, Lines 390–391) to enhance readability and clarity. We appreciate the reviewer’s careful review and valuable suggestion.
Reviewer comment 10: When discussing clinical trials, please provide more detailed information on the treatment modalities for IL-10 derivatives. Specifically, the number of injections, frequency, dose ranges, when available etc. For example, trials involving PEGylated forms of IL-10 could be explored in more detail. It should be possible to collect and present additional data regarding these aspects, which would improve the practical understanding of IL-10's therapeutic potential or failure
Response: We sincerely appreciate the reviewer’s insightful comment. In response, we have expanded the discussion on clinical trials to provide more detailed information on the treatment modalities of PEG-IL-10. Specifically, we have included details regarding patient randomization, number of injections, dosing frequency, dose ranges, combination treatments, and clinical outcomes in the revised manuscript (Page 11, Lines 462–514). We appreciate the reviewer’s valuable feedback, which has helped strengthen the manuscript: “Despite the encouraging outcomes reported above in the IVY study, pegilodecakin failed to demonstrate promising results in the subsequent CYPRESS 1 and 2 clinical trials (NCT03382899 NCT03382912). These randomized Phase II studies investigated whether the addition of pegilodecakin to immune checkpoint inhibitors (ICIs; anti-PD-1 monoclonal antibodies: pembrolizumab or nivolumab) offered therapeutic benefits for patients with metastatic NSCLC compared to ICIs alone [124]. Specifically, CYPRESS 1 enrolled patients with high PD-L1 expression (tumor proportion score [TPS] ≥50%), while CYPRESS 2 included patients with low or negative PD-L1 expression (TPS 0–49%) [124]. In CYPRESS 1, patients received either pembrolizumab monotherapy (N = 50) or a combination of pegilodecakin and pembrolizumab (N = 51), while in CYPRESS 2, treatments included nivolumab alone (N = 25) or combined with pegilodecakin (N = 27). Pegilodecakin was self-administered subcutaneously by the patents at fixed doses of 0.4 mg (for patients ≤80 kg) or 0.8 mg (for patients >80 kg) on days 1–5 and 8–12, with rest periods on days 6–7 and 13–14, as guided by the phase I IVY study findings. Overall, the pegilodecakin + anti-PD-1 combination therapy did not significantly improve ORR, PFS, or OS compared to anti-PD-1 monotherapy. For CYPRESS 1 (pegilodecakin + pembrolizumab vs. pembrolizumab alone), the results were: ORR (47% vs. 44%), PFS (6.3 vs. 6.1 months), and OS (16.3 months vs. not reached). For CYPRESS 2 (pegilodecakin + nivolumab vs. nivolumab alone), the results were: ORR (15% vs. 12%), PFS (1.9 vs. 1.9 months), and OS (6.7 vs. 10.7 months). Furthermore, treatment discontinuation rates due to severe hematologic AEs were doubled in the combination treatment arms of both studies [124].
In a constant line, the SEQUOIA study (NCT02923921), a global randomized phase III trial, assessed the efficacy and safety of adding pegilodecakin to the chemotherapy regimen of folinic acid, fluorouracil, and oxaliplatin (FOLFOX) as a second-line treatment for gemcitabine-refractory metastatic pancreatic ductal adenocarcinoma (PDAC). Patients were randomized to receive pegilodecakin + FOLFOX (n=283) or FOLFOX alone (n=284). Alongside the standardized FOLFOX regimen, pegilodecakin was self-administered subcutaneously at fixed doses as described above. Although the pegilodecakin + FOLFOX combination initially demonstrated enhanced antitumor activity compared to FOLFOX alone—indicated by immunostimulatory signals of the IL-10R pathway, including increased levels of IFN-γ and granzyme B and decreased TGF-β levels—its overall results showed no improvement in OS (5.8 vs. 6.3 months), PFS (2.1 vs. 2.1 months), and ORR (4.6% vs. 5.6%) [125,126]. Moreover, the incidence of grade ≥3 hematological AEs, particularly anemia, thrombocytopenia, and neutropenia, was notably higher in the pegilodecakin + FOLFOX arm, necessitating greater use of red blood cell transfusions and erythropoiesis-stimulating agents (5.8% vs. 2.0%). These findings underscore the need for further translational research to identify patients likely to benefit from pegilodecakin-chemotherapy combination in treatment-refractory advanced tumors [125,126].
Taken together, while the PEGylation process of recombinant human IL-10 (pegilodecakin) has notably enhanced its stability, serum concentration, and half-life, all while maintaining its biological activity, its overall benefits in the CYPRESS and SEQUOIA clinical trials has exhibited significant limitations [124-126]. The heterogeneity in cancer types, patient populations, and therapeutic combination types and regimens might complicate direct comparisons among these trials [58]. Importantly, although PEGylation is clinically approved for IFNα2a and IL-2 therapy and is currently under evaluation for additional immune-cytokine therapies such as IL-10 and IL-15, it is often associated with unwanted dose-dependent effects on bioactivity, primarily due to its amphiphilic nature [67]. In support, PEG-IL10-induced dose-dependent toxicity has been reported in both preclinical and clinical studies [53]. To overcome these challenges, further investigations, including the development of alternative and reprogrammed strategies optimizing IL-10 delivery and intratumoral localization, are warranted. These efforts are essential to fully harness and realize the mono- or combination-therapeutic potential of IL-10-directed therapy across a range of malignancies.”.
Respected Reviewer, once again, we sincerely thank you for your time and consideration, as well as for the opportunity to revise our manuscript. We deeply appreciate your valuable feedback and insightful comments, which have been instrumental in refining and strengthening our work.
Reviewer 2 Report
Comments and Suggestions for Authors
This manuscript reviewed “IL-10-Directed in Cancer Immunotherapy: Preclinical Advances, Clinical Insights, and Future Perspectives”. Basically, this rereview is timely and important. Revisions are suggested based on the questions as follows.
- “However, despite encouraging early research, 19 clinical trials of IL-10-based treatments, such as pegilodecakin, have faced challenges like side effects and inconsistent results, highlighting the need for better dosing and delivery methods.” It is suggested to summarize the “inconsistent results” in more detail, and to explain the possible mechanisms.
- In order to improve the readability of readers, it is recommended to add some pictures.
Author Response
Reviewer #2 (Comments and Suggestions for Authors):
This manuscript reviewed “IL-10-Directed in Cancer Immunotherapy: Preclinical Advances, Clinical Insights, and Future Perspectives”. Basically, this rereview is timely and important. Revisions are suggested based on the questions as follows.
- “However, despite encouraging early research, 19 clinical trials of IL-10-based treatments, such as pegilodecakin, have faced challenges like side effects and inconsistent results, highlighting the need for better dosing and delivery methods.” It is suggested to summarize the “inconsistent results” in more detail, and to explain the possible mechanisms.
- In order to improve the readability of readers, it is recommended to add some pictures.
Response: We sincerely appreciate the reviewer’s thoughtful feedback and valuable suggestion. In response, we have expanded the discussion on clinical trials in the revised manuscript (Page 11, Lines 462–514). We appreciate the reviewer’s valuable feedback, which has helped strengthen the manuscript: “Despite the encouraging outcomes reported above in the IVY study, pegilodecakin failed to demonstrate promising results in the subsequent CYPRESS 1 and 2 clinical trials (NCT03382899 NCT03382912). These randomized Phase II studies investigated whether the addition of pegilodecakin to immune checkpoint inhibitors (ICIs; anti-PD-1 monoclonal antibodies: pembrolizumab or nivolumab) offered therapeutic benefits for patients with metastatic NSCLC compared to ICIs alone [124]. Specifically, CYPRESS 1 enrolled patients with high PD-L1 expression (tumor proportion score [TPS] ≥50%), while CYPRESS 2 included patients with low or negative PD-L1 expression (TPS 0–49%) [124]. In CYPRESS 1, patients received either pembrolizumab monotherapy (N = 50) or a combination of pegilodecakin and pembrolizumab (N = 51), while in CYPRESS 2, treatments included nivolumab alone (N = 25) or combined with pegilodecakin (N = 27). Pegilodecakin was self-administered subcutaneously by the patents at fixed doses of 0.4 mg (for patients ≤80 kg) or 0.8 mg (for patients >80 kg) on days 1–5 and 8–12, with rest periods on days 6–7 and 13–14, as guided by the phase I IVY study findings. Overall, the pegilodecakin + anti-PD-1 combination therapy did not significantly improve ORR, PFS, or OS compared to anti-PD-1 monotherapy. For CYPRESS 1 (pegilodecakin + pembrolizumab vs. pembrolizumab alone), the results were: ORR (47% vs. 44%), PFS (6.3 vs. 6.1 months), and OS (16.3 months vs. not reached). For CYPRESS 2 (pegilodecakin + nivolumab vs. nivolumab alone), the results were: ORR (15% vs. 12%), PFS (1.9 vs. 1.9 months), and OS (6.7 vs. 10.7 months). Furthermore, treatment discontinuation rates due to severe hematologic AEs were doubled in the combination treatment arms of both studies [124].
In a constant line, the SEQUOIA study (NCT02923921), a global randomized phase III trial, assessed the efficacy and safety of adding pegilodecakin to the chemotherapy regimen of folinic acid, fluorouracil, and oxaliplatin (FOLFOX) as a second-line treatment for gemcitabine-refractory metastatic pancreatic ductal adenocarcinoma (PDAC). Patients were randomized to receive pegilodecakin + FOLFOX (n=283) or FOLFOX alone (n=284). Alongside the standardized FOLFOX regimen, pegilodecakin was self-administered subcutaneously at fixed doses as described above. Although the pegilodecakin + FOLFOX combination initially demonstrated enhanced antitumor activity compared to FOLFOX alone—indicated by immunostimulatory signals of the IL-10R pathway, including increased levels of IFN-γ and granzyme B and decreased TGF-β levels—its overall results showed no improvement in OS (5.8 vs. 6.3 months), PFS (2.1 vs. 2.1 months), and ORR (4.6% vs. 5.6%) [125,126]. Moreover, the incidence of grade ≥3 hematological AEs, particularly anemia, thrombocytopenia, and neutropenia, was notably higher in the pegilodecakin + FOLFOX arm, necessitating greater use of red blood cell transfusions and erythropoiesis-stimulating agents (5.8% vs. 2.0%). These findings underscore the need for further translational research to identify patients likely to benefit from pegilodecakin-chemotherapy combination in treatment-refractory advanced tumors [125,126].
Taken together, while the PEGylation process of recombinant human IL-10 (pegilodecakin) has notably enhanced its stability, serum concentration, and half-life, all while maintaining its biological activity, its overall benefits in the CYPRESS and SEQUOIA clinical trials has exhibited significant limitations [124-126]. The heterogeneity in cancer types, patient populations, and therapeutic combination types and regimens might complicate direct comparisons among these trials [58]. Importantly, although PEGylation is clinically approved for IFNα2a and IL-2 therapy and is currently under evaluation for additional immune-cytokine therapies such as IL-10 and IL-15, it is often associated with unwanted dose-dependent effects on bioactivity, primarily due to its amphiphilic nature [67]. In support, PEG-IL10-induced dose-dependent toxicity has been reported in both preclinical and clinical studies [53]. To overcome these challenges, further investigations, including the development of alternative and reprogrammed strategies optimizing IL-10 delivery and intratumoral localization, are warranted. These efforts are essential to fully harness and realize the mono- or combination-therapeutic potential of IL-10-directed therapy across a range of malignancies.”.
Respected Reviewer, once again, we sincerely thank you for your time and consideration, as well as for the opportunity to revise our manuscript. We deeply appreciate your valuable feedback and insightful comments, which have been instrumental in refining and strengthening our work.

Reviewer 3 Report
Comments and Suggestions for Authors
Formatting of Table 2 requires revision.
At least to me, the clinical findings are disappointing, given the promise suggested from preclinical research regarding the use of IL-10 as a potential therapeutic intervention.
Author Response
Reviewer #3 (Comments and Suggestions for Authors):
Comments and Suggestions for Authors
Formatting of Table 2 requires revision.
At least to me, the clinical findings are disappointing, given the promise suggested from preclinical research regarding the use of IL-10 as a potential therapeutic intervention.
Response: We sincerely appreciate the reviewer’s thoughtful feedback and valuable suggestions.
In response to the formatting concern, we have revised and improved the presentation of “Table 2” in the revised manuscript (Page 10, Lines 390–391) to enhance readability and clarity. We appreciate the reviewer’s careful review and attention to detail.
We also acknowledge and understand the reviewer’s perspective regarding the disparity between the promising preclinical findings and the relatively disappointing clinical outcomes. This discrepancy may be attributed to two key factors:
- The clinical trials primarily evaluated a single form of IL-10 preparation (PEGylated IL-10), which was administered systemically (subcutaneously).
- Recent advancements in preclinical research emphasize the intratumoral delivery of IL-10 (or its biologically active isoforms), specifically designed to interact with critical components of the tumor microenvironment. This targeted approach has demonstrated greater potential in mediating multimodal antitumor activity within tumor tissues while simultaneously reducing systemic adverse effects and minimizing inhibitory impacts on non-target cell types.
In alignment with this insightful comment, we have incorporated a detailed discussion addressing these aspects into the revised manuscript (Page 12, Lines 500–514). “Taken together, while the PEGylation process of recombinant human IL-10 (pegilodecakin) has notably enhanced its stability, serum concentration, and half-life, all while maintaining its biological activity, its overall benefits in the CYPRESS and SEQUOIA clinical trials has exhibited significant limitations [124-126]. The heterogeneity in cancer types, patient populations, and therapeutic combination types and regimens might complicate direct comparisons among these trials [58]. Importantly, although PEGylation is clinically approved for IFNα2a and IL-2 therapy and is currently under evaluation for additional immune-cytokine therapies such as IL-10 and IL-15, it is often associated with unwanted dose-dependent effects on bioactivity, primarily due to its amphiphilic nature [67]. In support, PEG-IL10-induced dose-dependent toxicity has been reported in both preclinical and clinical studies [53]. To overcome these challenges, further investigations, including the development of alternative and reprogrammed strategies optimizing IL-10 delivery and intratumoral localization, are warranted. These efforts are essential to fully harness and realize the mono- or combination-therapeutic potential of IL-10-directed therapy across a range of malignancies.”
Respected Reviewer, once again, we sincerely thank you for your time and consideration, as well as for the opportunity to revise our manuscript. We deeply appreciate your valuable feedback and insightful comments, which have been instrumental in refining and strengthening our work.

Round 2
Reviewer 1 Report
Comments and Suggestions for Authors
The authors have addressed almost all my concerns and improved the content and clarity of the manuscript for the benefit of readers.